# Hardware Acceleration and Implementation of YOLOX-s for On-Orbit FPGA

Ling Wang [1,2] , Hai Zhou [1,*], Chunjiang Bian [1], Kangning Jiang [1,2] and Xiaolei Cheng [1,2]

1   National Space Science Center, Chinese Academy of Sciences, Beijing 100190, China
2   University of Chinese Academy of Sciences, Beijing 100049, China
*   Correspondence: zhouhai@nssc.ac.cn

**Abstract:** The rapid development of remote sensing technology has brought about a sharp increase in the amount of remote sensing image data. However, due to the satellite's limited hardware resources, space, and power consumption constraints, it is difficult to process massive remote sensing images efficiently and robustly using the traditional remote sensing image processing methods. Additionally, the task of satellite-to-ground target detection has higher requirements for speed and accuracy under the conditions of more and more remote sensing data. To solve these problems, this paper proposes an extremely efficient and reliable acceleration architecture for forward inference of the YOLOX-s detection network an on-orbit FPGA. Considering the limited onboard resources, the design strategy of the parallel loop unrolling of the input channels and output channels is adopted to build the largest DSP computing array to ensure a reliable and full utilization of the limited computing resources, thus reducing the inference delay of the entire network. Meanwhile, a three-path cache queue and a small-scale cascaded pooling array are designed, which maximize the reuse of on-chip cache data, effectively reduce the bandwidth bottleneck of the external memory, and ensure an efficient computing of the entire computing array. The experimental results show that at the 200 MHz operating frequency of the VC709, the overall inference performance of the FPGA acceleration can reach 399.62 GOPS, the peak performance can reach 408.4 GOPS, and the overall computing efficiency of the DSP array can reach 97.56%. Compared with the previous work, our architecture design further improves the computing efficiency under limited hardware resources.

**Keywords:** convolutional neural network; remote sensing image processing; on-orbit high-performance computing; YOLOX-s; FPGA hardware acceleration



## 1. Introduction

The processing flow of traditional remote sensing image target detection is usually limited by the data transmission bandwidth, detection robustness, and real-time detection performance. Therefore, exploring high-efficiency edge computing has become a new solution to remote sensing image processing [1]. However, under the circumstance that onboard hardware resources and power consumption are severely limited, it is difficult to process massive remote sensing image data in real time, accurately, and efficiently, which makes image processing on satellites a challenge at present [2]. Object detection algorithms have developed rapidly since convolutional neural networks (CNNs) were used for object detection [3–8]. Among them, the YOLO [9–12] series of algorithms achieve a better detection speed by directly regressing and predicting the target position and category while ensuring the detection accuracy, which is more suitable for edge computing equipment with higher real-time requirements.

The field-programmable gate array (FPGA) has the advantages of a small size, a low power consumption, and reconfigurability, and it has received more and more attention in the direction of accelerating CNNs [13]. Compared with GPUs with a high power consumption and ASICs with long development cycles, FPGAs are more suitable for

the hardware acceleration of neural network models on satellite platforms with limited resources, space, and power consumption [14]. The performance of FPGA acceleration is closely related to the on-chip resources of hardware and different CNN structures. How to use limited hardware resources to design an accelerated processing architecture with a high efficiency and high performance on satellites is a crucial research topic today [15].

Currently, there are few studies on the use of an FPGA for the real-time inference of on-orbit target detection tasks. In the hardware acceleration research on the same series of YOLO networks, different hardware design architectures have been proposed for specific application scenarios. For example, Zhang et al. [16] proposed a low-latency accelerator architecture based on an FPGA-based dual-symbol multiplication correction circuit for the YOLOv2-tiny network model, but the DSP usage is not very high, resulting in a waste of limited resources. Bi et al. [17] designed a special floating-point matrix multiplication unit and double-buffered data processing circuit for the YOLOv2 algorithm to improve the calculation speed of the CNN, but the operating frequency of the accelerator architecture is low. Zhang et al. [18] developed a reconfigurable CNN accelerator based on an ARM + FPGA heterogeneous architecture, which uses the FPGA to accelerate operations in specific convolutional layers, but the inference performance of this architecture can be further improved. Nguyen et al. [19] presented an efficient implementation of the binary quantization network YOLOv2-tiny, which reduces the inference delay, but there is a greater loss in the detection accuracy of the network model. None of the above-mentioned designs considers the triple modular redundancy (TMR) to ensure the reliability of the calculation, and these designs are difficult to adapt to the complex particle radiation environment in space [20].

YOLOX [21] is an advanced target detection algorithm at present, which has a faster inference speed and a higher detection accuracy in the same series YOLO network and other single-stage detection networks such as EfficientDet in the public dataset COCO. Meanwhile, YOLOX's network design is regular, which can be compatible with the hardware architecture design of models of different sizes and has special advantages for satellite systems with a higher real-time inference performance and detection accuracy. Therefore, to satisfy the requirements for the target detection on satellites with high-density computing and an advanced detection performance, the main research work in this paper is as follows:

1.  Based on a detailed analysis of the YOLOX-s model, an efficient, low-latency, and reliable parallel processing architecture is proposed, which is fully compatible with YOLOX-s, YOLOX-l, and YOLOX-Darknet53 in the YOLOX family.
2.  The main modules in YOLOX are designed, and the core convolution module is further optimized. Meanwhile, the design strategy of a three-path prefetch cache queue is proposed to maximize the reuse pattern of on-chip data, relieve the bandwidth bottleneck of the external DDR in the multi-level cache, ensure the efficient operation of the DSP array, and improve the overall inference performance. The peak performance can reach 408.4 GOPS under the condition of a guaranteed detection accuracy.
3.  The design strategy of multi-layer small-scale cascade computing is adopted for the core pooling module and the three-path cache queue design reusing mode. In this way, the consumption of resources is reduced, and the logic timing is optimized.
4.  The real-time processing performance of the architecture was simulated and verified, and it was comprehensively analyzed from all aspects. Meanwhile, the related work of other FPGA-accelerated YOLO series networks was taken for comparison and analysis, which provides new research directions for the following research work.

## 2. Architecture Design and Module Implementation

### 2.1. Analysis of YOLOX-s

The design of the FPGA acceleration architecture is closely related to the specific structure of the YOLOX-s network. To ensure that the overall architecture is efficient, reasonable, and compatible, the structure of the YOLOX-s network is first introduced and analyzed in detail. There are five standard network structures in the YOLOX series of

network models, among which the four network models of YOLOX-s, YOLOX-m, YOLOX-l, and YOLOX have the same principle. They differ in the width and depth of the network structure, and the computation amount increases sequentially. The network structure of YOLOX-s is shown in Figure 1, which is mainly composed of four parts: the input, the backbone network, the neck network, and the prediction output.

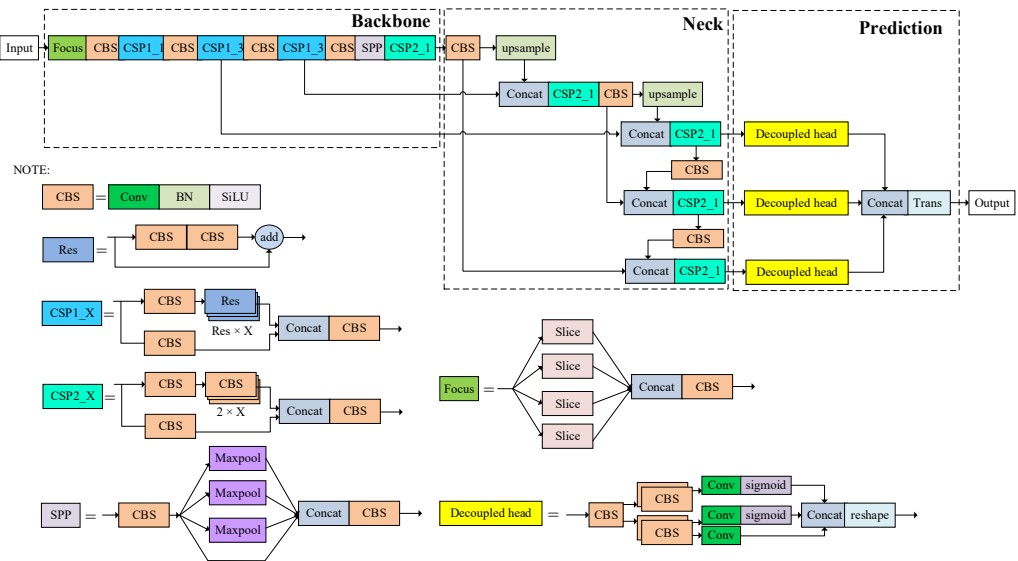

**Figure 1.** YOLOX-s network structure.

YOLOX-s contains many types of components, but the basic operation unit is composed of a large number of convolution layers. As shown in Figure 1, almost every convolutional layer is followed by batch normalization (BN) layers and Sigmoid-weighted linear unit (SiLU) layers to form CBS components. The focus component is a novel structure mainly composed of Slice operations, which can reduce the size of the feature map without losing the feature map information. There are two types of cross-stage partial (CSP) components, both of which use a residual (Res) structure, whereas the CSP1_X structure also integrates a separate residual component. In the YOLOX-s network structure, a convolutional layer with a stride of two is used directly instead of the traditional pooling layer for down-sampling. Only max-pooling layers with a stride of one and sizes of $5 \times 5$, $9 \times 9$, and $13 \times 13$ are used in the Spatial Pyramid Pooling (SPP) component to extract the local response features of the feature maps.

The network structure of YOLOX-s is relatively deep, e.g., the convolutional layer has 83 layers. The analysis of the network model structure shows that the convolutional layer occupies more than 90% of the operations of the entire neural network, and the number of operations reaches 24.85 GOPs, which occupies a large part of the forward inference computation time. Therefore, the inference efficiency of the overall architecture is determined by the computational efficiency of the convolution operation. Since the weight shapes of YOLOX-s and the other three network models of YOLOX are very regular, the number of input channels and the number of output channels of almost all convolution kernels have the same common factor. The hardware computing unit of an FPGA has specific implications for the design of an efficient computational array parallelism. In addition to the convolution, other calculations mainly occur in the pooling, residual, and activation functions. Therefore, to ensure the inference performance of the overall network, this paper optimizes each module, which will be described in the following architecture analysis and module implementation.

### 2.2. The Overall Architecture Design

In the structural analysis of YOLOX-s, it is mentioned that the main calculation of forward inference occurs in modules such as convolution and pooling, and the convolution

operation occupies almost all the computation of the YOLOX-s network model. Therefore, to improve the computational efficiency and reliability of YOLOX-s for the forward inference, this paper adopts different design optimization strategies for different modules, and the overall architecture of the FPGA-based hardware accelerator is shown in Figure 2.

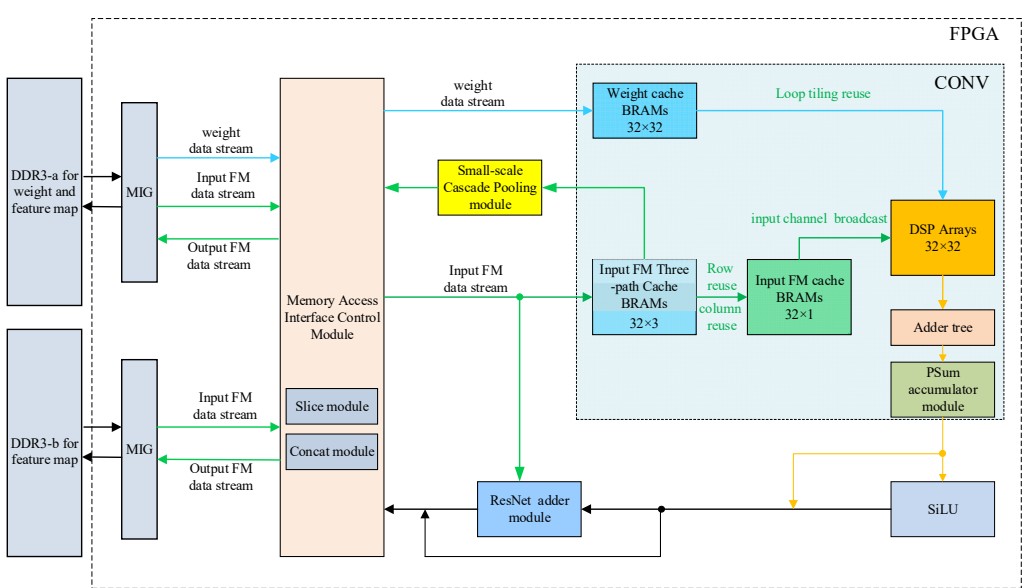

**Figure 2.** The overall architecture of the accelerator.

Due to the limited storage resources on the FPGA chip, the data volume of the feature maps and weight parameters is too large to be all cached on the FPGA, so this paper builds a multi-level cache structure. The external storage DDR3 is mainly used to store the weights of the model, as well as the feature map data. Based on different computing operations, specific optimization designs are conducted for the storage of the feature map data and weight data, including the optimization of the storage location and the storage data stream structure. For the optimization of the storage location, the feature maps used for residual calculation and data concatenating are stored in two external DDR3s, mainly to make full use of the data bandwidth of the two DDR3s to ensure efficient operations of the DSP array. The optimization of the storage data stream structure is based on the strategy of maximizing the on-chip data reuse mode and minimizing the partial sum cache, and the three-path cache queue design and data partitioning are adopted for implementation. BRAM is an on-chip cache resource. In the multi-level cache structure, BRAM is mainly used to build a cache queue, and data interactions with the DSP array are directly performed. For the core convolution module, to fully utilize the limited hardware resources while ensuring reliability, this paper designs a 32 × 32 DSP computing array in combination with the structure of YOLOX-s. Then, to overcome the bandwidth bottleneck of the external DDR3, this paper proposes the three-path cache queue to maximize the reuse of on-chip data, thereby ensuring an efficient parallel computing of the DSP array and improving the overall network inference performance. For the pooling module, this paper proposes the small-scale cascaded pooling computation strategy and designs a four-level comparison array. This realizes the reuse of the three-path cache queue design, enhances the reuse of feature maps, further reduces the design complexity, and saves on-chip resources.

The workflow of the accelerator is as follows: first, the FPGA controls the Memory Interface Generator (MIG) core to read part of the input feature map (Input FM in Figure 2) and weight from the external DDR memory through the memory access interface control module and caches the input feature map in the three-path cache queue, and the weight is cached in the weight queue. Then, according to the optimized loop calculation sequence, combined with row reusing and column reusing, one-path input feature map data is selected from the three-path cache queue and sent to the DSP array with the weight data for

calculation. After the calculation is completed, the partial sum (PSum) of the output feature map is obtained through the five-level adder tree. The convolution result can be directly obtained after the partial sum passes through the PSum accumulator module. Depending on whether it needs to go through the SiLU function, different data flow directions are chosen for the residual addition operation. Finally, the MIG core is controlled by the memory access interface control module to write the processed output feature map data back to the external DDR memory. For the small-scale cascade pooling module, the use of the three-path cache queue can realize a logic reuse similar to the DSP arrays in convolution operations, thus completing the calculation of the comparison arrays in the small-scale $3 \times 3$ module.

In addition to the convolution and pooling, the design and implementation of other modules will also be discussed in detail in the following subsection.

### 2.3. Design of Convolution Module

In the design of the overall architecture, considering that convolution occupies the core position of the overall design, to ensure that the convolution calculation can be performed as efficiently as possible, the following optimization strategies are adopted in this paper:

5. In the conditions allowed by the FPGA on-chip computing resources, according to the structural characteristics of the YOLOX-s, the strategy of multi-dimensional parallel loop unrolling is selected to build the largest parallel computing array and optimize the logic timing, thus improving the processing performance of the accelerator.
6. In the conditions allowed by the on-chip cache resources of the FPGA, the three-path cache queue design is proposed by adopting the strategy of maximizing on-chip cache data reuse. This overcomes the bandwidth bottleneck caused by the external DDR3 in the multi-level cache structure, thereby guaranteeing the computational efficiency of the DSP array. Meanwhile, the number of accesses to external storage is reduced as much as possible to reduce the hardware power consumption.
7. In the conditions allowed by the on-chip cache resources of the FPGA, an optimization strategy is adopted to minimize the partial sum cache. Combined with the three-path cache queue, data partitioning, and loop sequence optimization, the cache pressure of the data input queue and output partial sum is balanced, which further ensures the efficient calculation of the DSP array.

#### 2.3.1. Design of Parallel Loop Unrolling

According to the strategy of building the largest computing array based on parallel loop unrolling, the computing resources should be fully utilized so that the DSP array can perform an efficient computing. The convolution calculation is composed of four-dimensional loops, so loop unrolling can be divided into convolution kernel loop unrolling, input channel loop unrolling, input feature map loop unrolling, and output channel loop unrolling [22]. The ways and combinations of a different loop unrolling determine the design of the convolutional computing array, which further affects the memory access and data reuse patterns. Therefore, different loop unrolling dimensions need to be determined based on the structural characteristics of YOLOX-s.

The network model structure of YOLOX-s is relatively deep, and the number of convolutional layers reaches 83 layers. The size of the convolution kernel includes $3 \times 3$ and $1 \times 1$, of which the $1 \times 1$ convolution kernel is the majority. If convolution kernel loop unrolling is used for designing the convolution array, it will cause a great waste of resources. The analysis of the network parameters of YOLOX-s indicates that the greatest common factor of the number of input channels and the number of output channels of all the convolution layers except the first layer and the prediction structure is 32. Considering the number of DSPs in the FPGA and the size of the BRAM, to maximize the parallelism of the processing architecture and meet the requirements of the TMR, the YOLOX-s network adopts the parallel loop unrolling strategy for the input channels and output channels. The degree of parallelism for the loop unrolling of the input channel and output channel is both

32, so the overall parallelism of the hardware processing architecture is 1024. For network layers with input channels less than 32, this paper uses the methods of feature map filling and feature map reshaping to ensure that the DSP computing resources are fully utilized.

The FPGA used in this paper has a total of 3600 DSP units. The dimension of the input feature map loop unrolling is not considered because the architecture is designed for the satellite platform. Meanwhile, considering the complex space environment and the impact of space particle radiation on the stability of FPGAs [23], such as the single event effect (SEE) [24], the number of DSP units available in the FPGA in this design is only one-third of the total.

### 2.3.2. Convolution Computation Array Architecture

According to the optimization strategy of the parallel loop unrolling of the input channel and output channel, the parallel computing DSP matrix is designed as shown in Figure 3. Since the weight data needs to be updated for all the computing units at the same time from the cache queue, the weight cache queue is set to 32 × 32. As a result, there are 32 weight cache queue groups of convolution kernels, and each group contains 32 BRAM weight cache queues that are synchronized with the input channel data, with a total of 1024 weight cache queues. When updating the weight data, the weight cache queue group of each convolution kernel only updates the 32 calculation units of the current calculation matrix column, so the entire calculation matrix can update the weights of 1024 calculation units at a time. For the input feature map cache queue, the computing DSP array is updated through one-to-many column broadcasting. When updating the feature points, each input channel copies the current column feature points 32 times through the five-level pipeline and then fills them into the current row. The whole calculation matrix contains 32 input channels, so the input feature map updates 32 feature points each time, and each column has 32 feature points. After the weight data and the input feature points are updated, each computing unit will perform a multiplication operation on the weight data and the input feature data. Subsequently, each output channel in the column direction uses a five-level pipeline adder tree to add the 32 products in this channel to obtain a partial sum. Finally, the partial sum result is passed through the accumulator module to obtain the final result for the output.

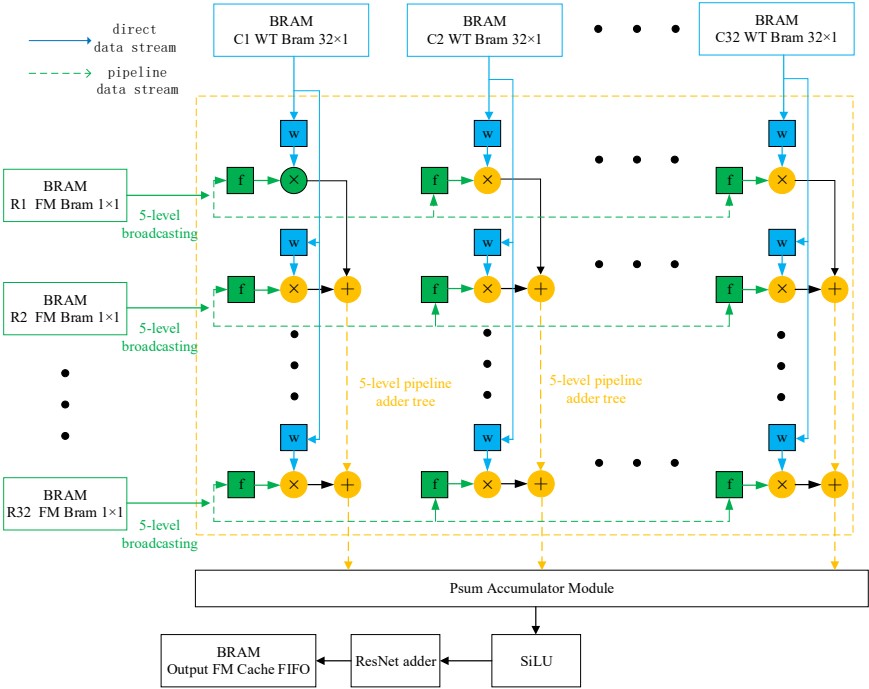

**Figure 3.** YOLOX-s convolution calculation array architecture.

### 2.3.3. Multi-Level Cache and Bandwidth Analysis

In the overall architecture, a multi-level caching strategy is adopted to better solve the problems of data storage and computing. To ensure that the DSP array can perform efficient convolution computations, a balance should be achieved between the bandwidth demanded by the DSP array and the bandwidth supplied by the external DDR. Therefore, the best theoretical performance of our overall architecture is determined by the design and optimization of the overall architecture based on the bandwidth analysis in the multi-level cache structure. The supply and demand relationship of the bandwidth in the multi-level cache structure is shown in Figure 4.

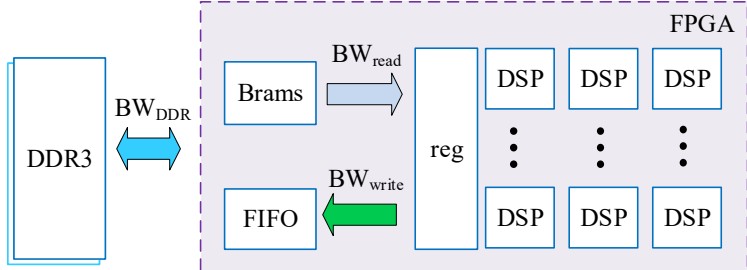

**Figure 4.** Bandwidth supply and demand relationship in the multi-level cache structure.

In the design of the overall computing architecture, combined with the hardware cost of aerospace-grade DDR3, weights and feature maps are stored in two DDR3s. The working frequency of the DDR3 memory is 800 MHz, and the data width is 64-bit. Experiments indicate that the DDR3 working efficiency under the MIG core is about 60–80%, so the bandwidth of a single-granular DDR3 memory is about 60–80 Gbps by Equation (1).

$$BW_{DDR3} = 2f \times d \times \eta \tag{1}$$

where $f$ is the working frequency, $d$ is the data width of the DDR3 bus, and $\eta$ is the working efficiency of DDR3 under MIG ($f = 800$ MHz, $d = 64$, $\eta = 0.7$).

The input of the DSP array is mainly the weight and feature map data stream. The data precision of the weight parameters and the feature map is 16-bit, and the working frequency is 200 MHz. Therefore, the calculation formula of the weight read bandwidth of the DSP array and the read and write bandwidth of the feature map is represented as follows:

$$BW_{Read} = \frac{f \times p \times \beta}{\gamma} \tag{2}$$

$$\gamma = reuse\ times \times \alpha \tag{3}$$

$$BW_{Write} = \frac{output\ channel \times BW_{Read}}{input\ channel \times kernel\ size \times kernel\ size} \tag{4}$$

where $f$ is the working frequency, $p$ is the data quantization precision, $\beta$ is the number of input data queues in the DSP array, $\gamma$ is the reuse degree of the feature map data in the on-chip cache, and $\alpha$ is the reuse coefficient in the spatial position, which depends on the size of the convolution kernel and the sliding stride (in this paper, $f = 200$ MHz, $p = $ fix16, $\beta = 32$ for FM or $\beta = 1024$ for weight, $reuse\ times = \frac{output\ channels}{32}$. $\gamma = \alpha = 1$, when there is no reuse).

The weight parameters and feature points are updated at the same time. When performing convolution operations, this paper adopts the strategy of a minimum partial sum cache to reduce the storage pressure brought by the partial sum. Therefore, 1024 weight parameters need to be updated per clock, and in a complete convolution operation, the weights are not reused. It can be obtained from the Formula (2) that the bandwidth requirement of the weight is as high as 3200 Gb/s, which far exceeds the bandwidth of DDR3, thus easily resulting in an extremely low computational efficiency

of the accelerator architecture. Combined with the network structure characteristics of YOLOX-s and the FPGA hardware resources, the number of weight parameters of each layer is not too large, so the weight of each layer can be completely stored in the on-chip cache of FPGA. Therefore, when the convolution layer starts to be calculated, our design reads all the weights of the current layer into the FPGA on-chip cache, and these weights can be reused continuously, thus avoiding the weight bandwidth bottleneck of the efficient operation and reducing the power consumption of frequent external memory accesses.

Due to a large number of parameters, the feature map cannot be completely stored in the on-chip cache, so multiple accesses to the DDR are required. In each clock cycle, 32 channels of the feature map need to be updated, and the read bandwidth is closely related to its reuse degree. When there is no reuse, the bandwidth is 100 Gb/s according to Formula (2), which is greater than the bandwidth of DDR3, but when the reuse degree $\gamma$ is higher, the bandwidth requirement is lower.

For a $1 \times 1$ convolution operation, the reuse degree $\gamma$ is the reuse times of the loop tiling ($\alpha = 1$), and the reuse times is the ratio of the number of output channels to the degree of parallelism of the output channels. For the $3 \times 3$ convolution operation, the reuse degree $\gamma$ is dependent on the reuse times and is related to the reuse coefficient $\alpha$. When the step stride is different, the spatial reuse coefficient $\alpha$ is also different, as shown in Figure 5. In a complete convolution operation, when the stride is one, the reuse coefficient $\alpha$ can reach a maximum of nine, and the minimum is four. When the stride is two, the maximum can be 25/9, and the minimum can be 16/9. In addition, for a $3 \times 3$ convolution operation, it can be seen from Equation (4) that the write bandwidth requirement is also greatly reduced compared to the read bandwidth, so it will not cause a bandwidth bottleneck.

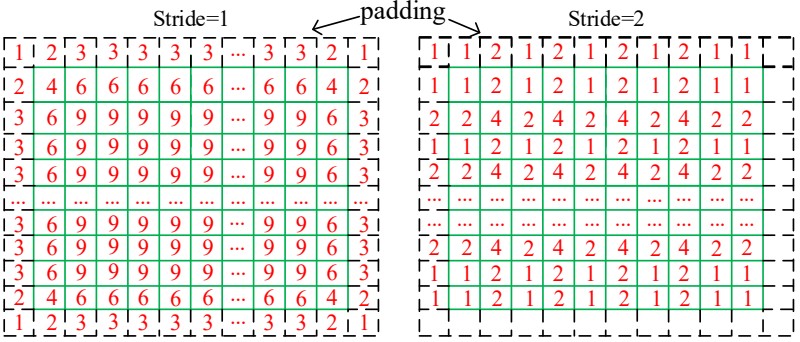

The number in the box is the reuse coefficient $\alpha$ at the current position (including the input channel direction).

**Figure 5.** $3 \times 3$ convolution reuse coefficient $\alpha$ under different strides.

To sum up, only the convolutional layer with a convolution kernel size of $1 \times 1$ and an output channel number of 64 or fewer will cause a slight bandwidth bottleneck, and the DSP array will have a short reading wait. However, the proportion of these layers is very small, and the computational efficiency of the remaining convolutional layers can almost reach 100%, so it has only a small impact on the overall inference efficiency.

In fact, if conditions permit, only adding an additional DDR3 can overcome the bandwidth bottleneck (Formula (1)–(4)), and all layers can achieve a computing efficiency of over 99%. However, considering the high design costs of aerospace-grade DDR3 used on satellites, the current system design is better because of the small gains from the huge increase in hardware cost.

### 2.3.4. Data Partitioning and the Three-Path Cache Queue Design

Combined with the design of the multi-level cache structure, the complete feature map of each layer from the external memory DDR3 needs to be divided into multiple smaller data blocks, and then these blocks are read into the on-chip cache in sequence, i.e., data partitioning. In our design, combined with the address width $d$ and burst length $l$ of DDR3, and the data precision $p$ ($d = 64, l = 8, p = 16$), the data block of the input feature map is

divided along the direction of the input channel of the feature map, and the parallelism of the input channel is used as the basic unit, as shown in Figure 6. The data bands of different colors in the figure are the basic units of data that are divided. The block feature map data is beneficial to the data flow control of the DSP array input and makes it easier to perform loop tiling reuse and reuse in the spatial dimension of the three-path cache queue.

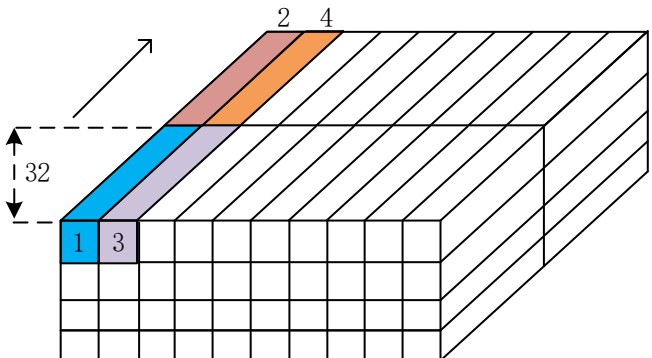

**Figure 6.** Input feature map data partitioning.

The weight data partitioning is based on the parallelism of the input channel and the parallelism of the output channel, and the block weight data is the basic unit for storage, calculation, and reuse. Figure 7 shows the data partitioning of the convolution kernel with a size of $64 \times 3 \times 3 \times 32$. Based on the bandwidth analysis of the above architecture, to ensure the highest computing efficiency of the DSP array, the weight data stream of each layer is fully cached after the weight data is divided into blocks by a strategy of caching while calculating. This maximizes the reuse efficiency of the feature maps, thereby avoiding the time and power consumption overhead caused by a frequent access to the external storage of weight data.

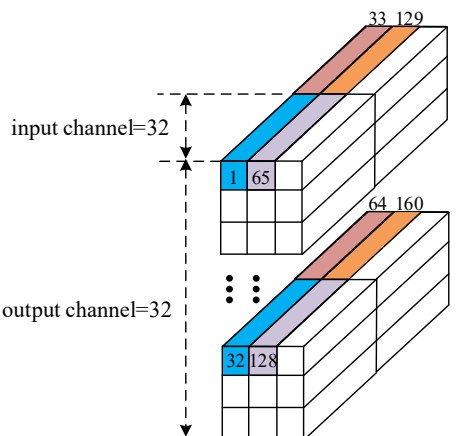

**Figure 7.** Weight data partitioning.

Through data storage and data flow optimization, feature map and weight data partitioning further enhance the feature map and weight reusing, and overcomes the DDR3 bandwidth limitations, thus ensuring an efficient operation of the DSP array. Based on data partitioning, this paper optimizes the loop calculation sequence of the DSP array by adopting a design strategy that minimizes the partial sums of the cache. Through the fastest approach to output the feature map, the cache pressure of the partial sum is reduced, and the balance of the BRAM resource utilization in the output and input queues is realized.

The sequential control flow of the loop calculation is shown in Figure 8. The input feature map data block is loaded from the external DDR memory to the three-path cache queue along the input channel, and the basic block unit of the weight is sequentially loaded from the external memory to the weight on-chip buffer. The convolution calculation

starts when the weights are loaded into the block data that can be used for the first convolution calculation.

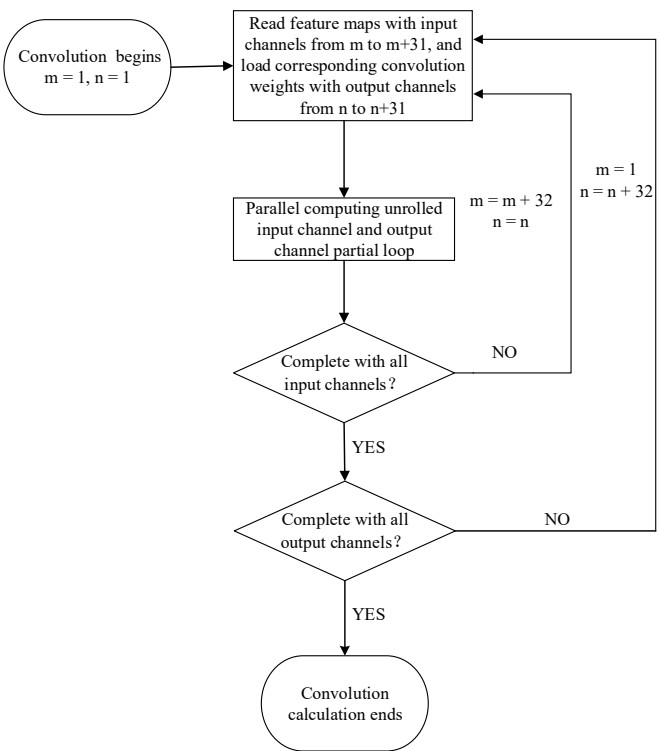

**Figure 8.** Flow chart of the loop calculation sequence.

The reuse of the spatial position of the feature map, which is introduced in the analysis of the bandwidth requirements, is mainly implemented through the three-path cache queue. As shown in Figure 9, the analysis is performed with a convolution with a size of $3 \times 3 \times 64$ and a stride of one. The three-path cache queue can prefetch feature maps and reuse the spatial position, based on the strategy of maximizing the reuse of feature maps and minimizing the partial sum cache. On the right side of Figure 9, the blue stripes represent the data reuse in the column direction in the cache queue, and the green stripes represent the data reuse in the row direction at the current position. The length of the blue and green strips is the depth of the data reuse, and its size can be calculated by the following Formula (5)–(6):

$$Col\ Reuse\ Length = \frac{(kernel\ size - stride) \times input\ channel}{32} \tag{5}$$

$$Row\ Reuse\ Length = \frac{FM\ width \times inpu\ channel}{32} \tag{6}$$

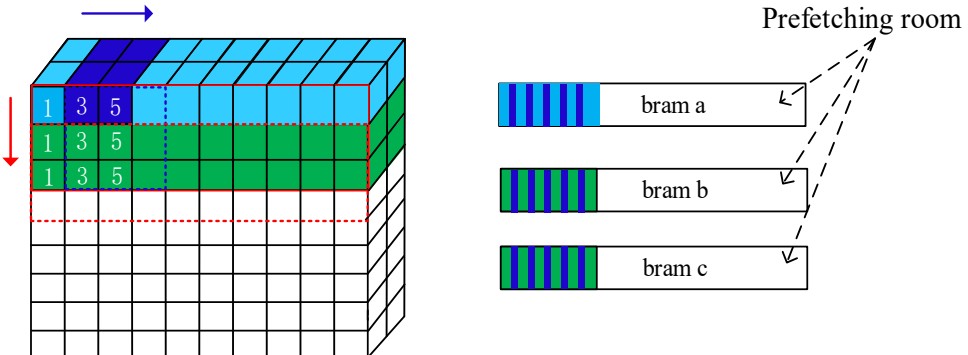

**Figure 9.** Schematic diagram of spatial reuse and the three-path cache queue design.

The design process of the three-path cache queue is shown in Figure 10, and the logic of the row reuse is illustrated in this figure. When the reuse flag of a queue is high, the data in the queue that has been loaded is reused at this time. With the operation of the DSP array, every two (stride = one) or one queue (stride = two) is alternately combined and reused in order. In the figure, the length of the arrow is the depth of the reuse in the row direction, which can be calculated by the Formula (6). Our design combines the three-path cache queue to maximize the reuse of the feature maps. When the sliding stride is one or two, the data reuse rate can reach 88.9% or 55.6%, respectively, and the highest reuse coefficient $\alpha$ can reach nine, which greatly reduces the theoretical bandwidth requirement of DDR3.

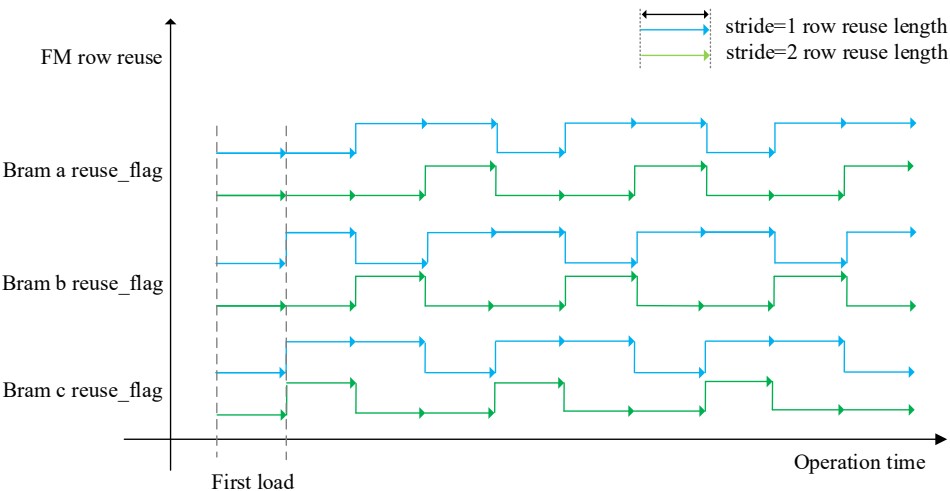

**Figure 10.** Schematic diagram of row reuse logic design of feature map.

### 2.4. Design of the Small-Scale Cascade Pooling Module

To ensure the inference performance of the overall architecture, this paper adopts an optimization strategy for small-scale cascaded computations in the pooling module of YOLOX-s. The YOLOX-s network model only uses max-pooling layers with a stride of one and sizes $5 \times 5$, $9 \times 9$, and $13 \times 13$ in the SPP structure. Since the pooling window size of these three max-pooling layers is large and the step size is one, if the comparator array is directly used for calculation, it will require a lot of time, and the data reuse is very low, which will cause a huge waste of resources. Therefore, instead of using the initial large-size pooling window, this paper uses a multi-layer small-scale cascade pooling method for the optimization of resources.

Taking $5 \times 5$ max-pooling as an example, as shown in Figure 11, if a $5 \times 5$ input feature map is max-pooled with a window of stride one and size $3 \times 3$, a new feature map will be obtained. If a $3 \times 3$ window is employed to perform a max pooling on the feature map again, the maximum value of the original $5 \times 5$ feature map can be obtained. This is equivalent to directly using a $5 \times 5$ window to operate the $5 \times 5$ feature map to take the maximum value.

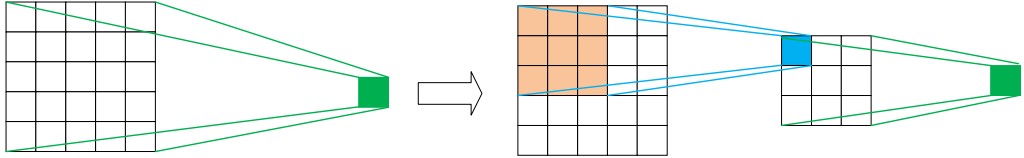

**Figure 11.** The schematic diagram of $5 \times 5$ multi-layer small-scale cascade pooling.

For the design of the $3 \times 3$ pooling module, a comparison array of four-level pipelines is used. The specific operation is shown in Figure 12. Combined with the three-path cache queue strategy, each queue caches a whole line of data of the input feature map, and

the data in the queue are loaded into the comparison array of $3 \times 3$ pooling windows in turn. The size of the four-level comparison array is $32 \times 3$, $32 \times 3$, $32 \times 1$, and $32 \times 1$, respectively. In Figure 12, each level of the light blue block represents the unit involved in the calculation, and the green block represents the final output. After the window is filled, the data in the first two columns of the window are compared, and the larger value obtained in each row is compared with the data in the third column of the window at the next level to obtain a $32 \times 3$ data column. Then, after two levels of comparison, the maximum value of $3 \times 3$ pooling is obtained. The design of the $3 \times 3$ pooling module makes full use of the three-path cache queue design strategy proposed in the convolutional array, thus realizing the maximum reuse of the feature maps. The data reusing logic of the $3 \times 3$ pooling operation is the same as the convolution operation, but the depth of the data reuse is different.

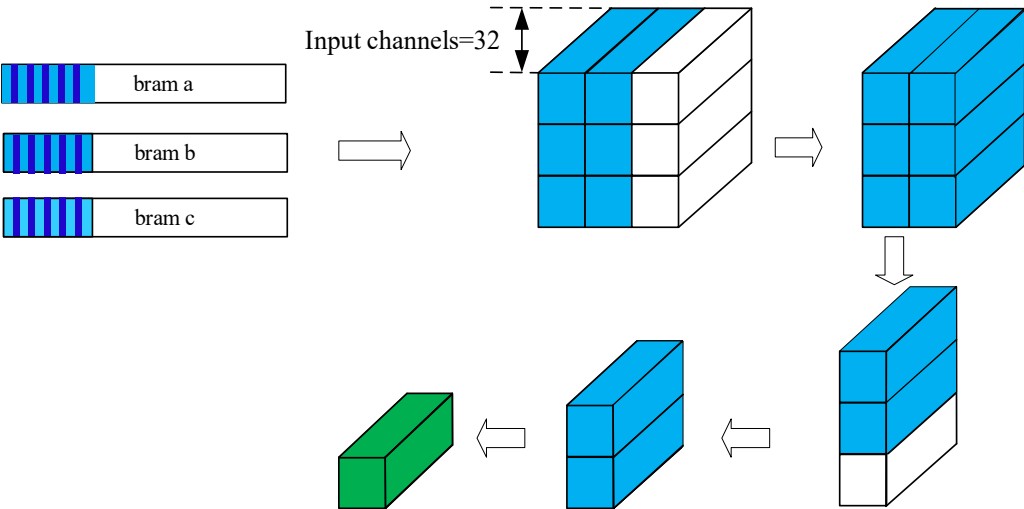

**Figure 12.** Schematic diagram of the $3 \times 3$ pooling operation.

For a single input feature map (input channel = 1), when a $5 \times 5$ pooling window is used for max-pooling, six-level pipelines are required to output the maximum value, and a total of 24 comparators are required. However, only 8 comparators are needed when max pooling is achieved by a two-level $3 \times 3$ pooling cascade. Similarly, for $9 \times 9$ and $13 \times 13$ pooling, if they are optimized with four-level and six-level $3 \times 3$ pooling, respectively, the number of comparators they require decreases from 80,168 to 8, respectively. By using multi-level $3 \times 3$ small-scale cascade pooling windows instead of original pooling windows, this optimization design method effectively reduces the complexity of the design, realizes the reuse of resources, and greatly reduces the cost of hardware resources.

### 2.5. Design of Slice Module

The Focus component of YOLOX-s is mainly composed of slice operations and convolution operations. The slice operation is a down-sampling operation that does not lose the feature map information, and its essence is not to calculate the feature map data. When a data stream comes in, it only needs to be stored in a specific storage structure.

Assuming that the size of the input feature map is $N \times N \times 3$, the operation of the slice module is shown in Figure 13. The counter $n$ is mainly used to judge whether the current row is odd or even, and the another counter $cnt$ is mainly used to count the batches of odd and even columns. In the strategy adopted here, along the input channel, odd-numbered columns of data are stored consecutively next to even-numbered columns, so the size of each slice count is 6 ($2 \times 3$). After each even-numbered row data slice is completed, since the odd-numbered row data is continuously stored along the input channel, before starting the odd-numbered row data splicing, the starting address needs to be updated to make it adapt to the previous even-numbered row.

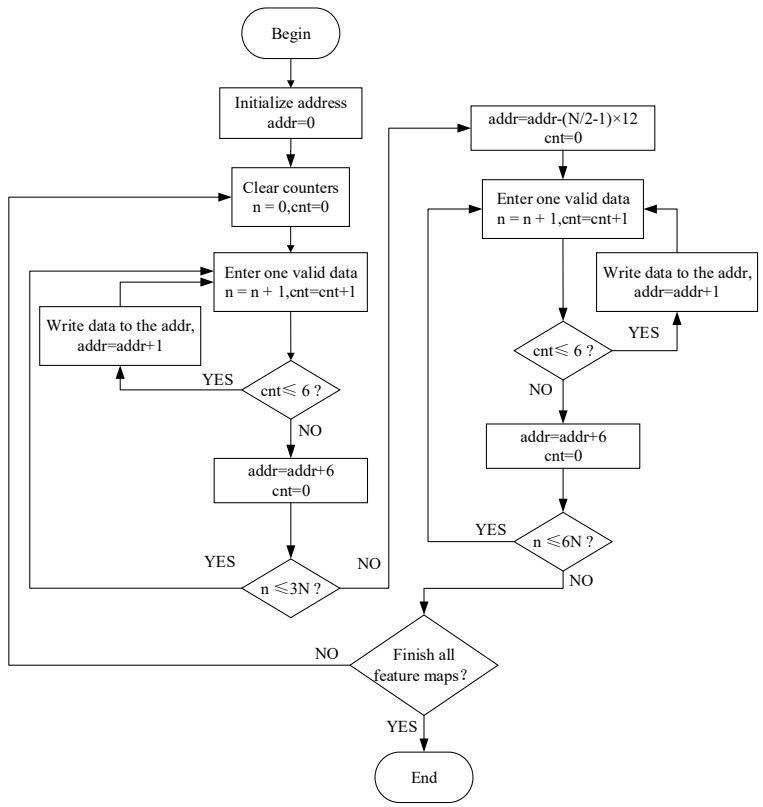

**Figure 13.** The flow chart of the slicing operation.

## 2.6. Design of Residual Module

The YOLOX-s network model uses residual components in the CSP structure. The residual component is composed of two convolutional layers and an adder layer. Therefore, the residual operation is essentially a convolution calculation, which is realized based on the convolution calculation module.

The structure of the residual calculation module is shown in Figure 14. It is mainly composed of the input feature map cache queue, the convolution acceleration module, and the residual addition module. The input feature map cache queue is a special prefetch queue opened up for the feature map data before the convolution calculation in the residual network. To prevent this part of the feature map from occupying the read bandwidth of the three-path cache queue, this paper adopts a separate storage strategy and stores this part of the feature map in another DDR3-a with weights. When the two-level convolution calculation is completed, the input feature map data in the cache queue will be added after the two convolutions are completed, thus obtaining the final output feature map.

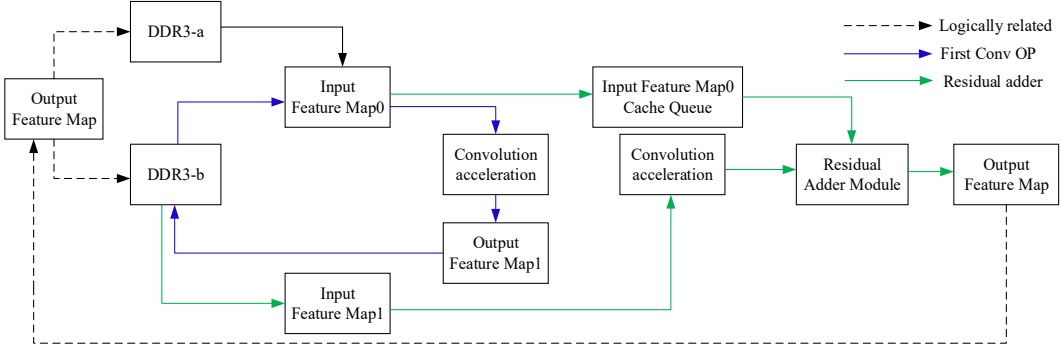

**Figure 14.** The schematic diagram of the residual calculation module.

### 3. Experiment

*3.1. Experimental Environment Setup*

In our experiments, the Xilinx Vivado 2020.1 integrated development environment was used for the design. The programming language is Verilog HDL, and Modelsim is used for the simulations. Meanwhile, the VC709 kit of Xilinx's Virtex-7 series is used as the basic implementation platform. It is equipped with an FPGA chip with a model of XC7VX690T. The chip crystal oscillator generates a clock frequency of 200 MHz, and there are 693,120 logic units, 52,920 Kb of BRAM, 3600 DSPs, etc. Two 4 Gb DDR external connections to FPGA, which is Micron MT8KTF51264HZ. The actual processing board is shown in Figure 15.

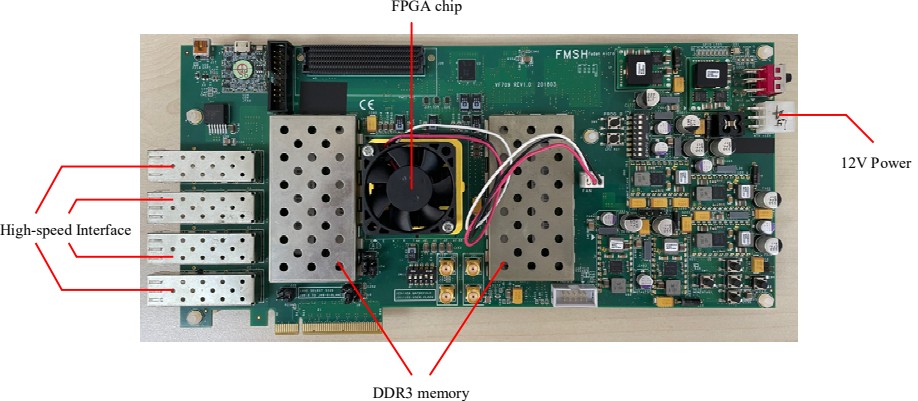

**Figure 15.** FPGA processing board for the experiment.

*3.2. Results and Comparison Analysis*

Considering the particularity of the satellite-to-ground target detection task, to ensure a target detection accuracy, the data format used is a 16-bit fixed point. The clock constraint is set to 200 Mhz, after Vivado synthesis and implementation, the highest operating frequency can reach 202.7 Mhz, and the FPGA resource usage is presented in Table 1. Although the FPGA used in the experiment has 3600 DSP units, considering the use of the platform on the subsequent satellite, TMR processing is required in a single FPGA, so the actual number of DSPs available is 1200. The total power consumption of the FPGA is 14.76 W, of which the static power consumption is only 0.672 W. Dynamic power consumption is mainly used for the signals, IO, clocks, BRAM, and logic, accounting for about 80% of all dynamic power consumption.

**Table 1.** FPGA resource usage.

| Resource | Used | Available | Utilization % |
|---|---|---|---|
| LUT | 165,981 | 433,200 | 38.32% |
| LUTRAM | 3329 | 174,200 | 1.91% |
| FF | 228,309 | 866,400 | 26.35% |
| BRAM | 1094 | 1470 | 74.42% |
| DSP | 1024 | 1200 | 85.33% |
| IO | 231 | 850 | 27.18% |
| BUFG | 8 | 32 | 25.00% |
| MMCM | 5 | 20 | 25.00% |
| PLL | 2 | 20 | 25.00% |

It can be seen from Table 1 that 1024 DSPs are used, and the utilization rate is as high as 85.33%. Since the throughput of the accelerator depends mainly on the number of DSPs used in addition to the quantization precision $p$ and operating frequency $f$, the higher the utilization rate of the DSP, the more the designed accelerator processing architecture can utilize the computing resources of the FPGA, and the better it can reflect the superiority

of the advanced architecture design. In addition, the BRAM also has a high resource utilization, which is mainly to optimize the timing of the processor architecture, thus increasing the reuse of on-chip data and minimizing the number of the FPGA accesses to an external DDR storage. The high usage of BRAM resources ensures that the proposed accelerator processing architecture does not block data while the pipeline is running, thus improving the overall operating frequency of the system.

Combined with the previous theoretical analysis of bandwidth (Section 2.3.3), the DSP computing efficiency of each convolutional layer is affected by the size of the convolution kernel, the number of output channels, and the number of input channels. The DSP computing efficiency of each layer is defined as the ratio of actual computing time to the ideal computing time in convolution computing. The ideal computing time is only the time when the DSP array is performing computations to complete the current convolution computation at every CLK. The actual computing time includes the time that data is loaded from the memory to the DSP array and written back to the memory from the DSP array, and the time it takes for the DSP to perform the computation. Therefore, the actual computing time is greater than the ideal computation time in any case.

To more clearly analyze and illustrate the computational efficiency of the overall architecture, the simulation waveforms of three different convolutional layers are analyzed. At the operating frequency of 200 MHz, the functional simulation waveform is shown in Figure 16, where the convolution kernel size is $128 \times 64 \times 3 \times 3$, the input feature map size is $64 \times 160 \times 160$, and the step size is 1. The ideal computing time is 9.216 ms. The actual time is 9.239 ms, and the DSP computing efficiency is up to 99.75%.

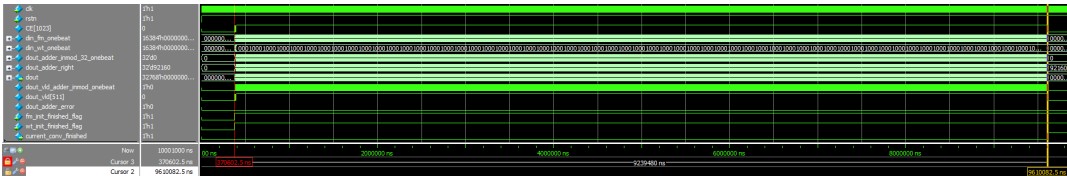

**Figure 16.** The simulation waveforms of $128 \times 64 \times 3 \times 3$.

For the convolution kernel size $64 \times 64 \times 1 \times 1$ and the input feature map size $64 \times 80 \times 80$, the functional simulation waveforms are shown in Figure 17. The ideal calculation time is 0.128 ms, while the actual time is 0.153 ms, so the calculation efficiency of the DSP is 83.6%.

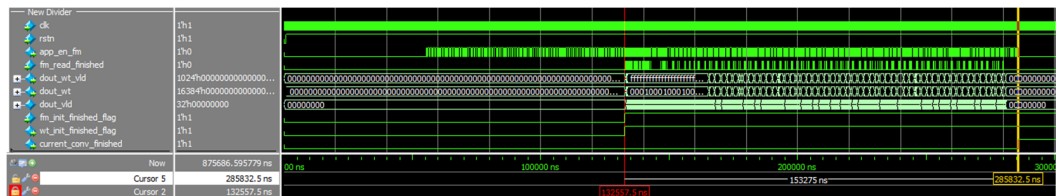

**Figure 17.** Simulation waveforms of $64 \times 64 \times 1 \times 1$.

For the convolution kernel size $32 \times 64 \times 1 \times 1$ and the input feature map size $64 \times 160 \times 160$, the functional simulation waveform is shown in Figure 18. The ideal calculation time is 0.256 ms, while the actual time is 0.463 ms, so the calculation efficiency of the DSP is 55.3%.

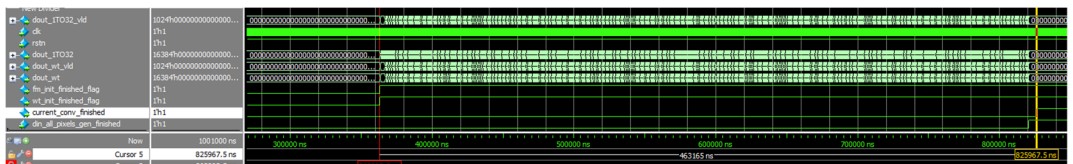

**Figure 18.** Simulation waveforms of $32 \times 64 \times 1 \times 1$.

Combined with the bandwidth analysis in Section 2.3.3, the simulation results in Figures 16–18 show that with the reduction in the output channels and the reduction in the convolution size, the reuse degree $\gamma$ will become smaller and smaller. For a small number of convolutional layers in the network, there will be a bandwidth bottleneck of DDR3. This article analyzes the inference results of all network layers in the YOLOX-s network and divides the computing efficiency of different convolutional layers into three categories according to the reuse degree $\gamma$, as shown in Table 2. The entire YOLOX-s network contains 83 convolutional layers, and the total inference time is 62.187 ms. The inference performance is 399.6 GOPS. From Equation (7) [16], it can be obtained that the DSP computing efficiency for the network inference is as high as 97.56%.

$$DSP_{efficiency} = \frac{Throughput}{\beta \times DSP_{used} \times f} \tag{7}$$

where $\beta$ is the operation that can be handled by one DSP in one cycle [25] ($\beta = 2$ in Fix-16, $\beta = 4$ in Fix-8, $DSP_{used} = 1024$, $f = 200$ MHz).

**Table 2.** The computational efficiency of different convolutional network layers.

| Categories | $\gamma$ | Layers/Total | Least Efficiency % |
|---|---|---|---|
| Kernel size = 1, Output channel $\leq$ 32 | 1 | 9/83 | 55.3 |
| Kernel size = 1, Output channel = 64 | 2 | 9/83 | 83.5 |
| Others | $\geq$3 | 65/83 | 99.1 |

Finally, our work is compared with other related work. As shown in Table 3, compared to [17,18], our work has an obvious advantage in both the throughput and the performance efficiency ratio. For [16], the reason for the slightly lower results in the throughput is that the first one uses 8-bit quantization. In this paper, due to the particularity of target detection tasks on satellites, 16-bit quantization is used to ensure the detection accuracy of the overall network. If the same data precision is adopted, our architecture will have greater advantages in terms of the throughput and performance efficiency ratio. In addition, our performance is also better than [16] in terms of the DSP computing efficiency, achieving a performance improvement of 2.36%.

**Table 3.** Comparison of different YOLO implementations on FPGA.

| Method | ISCAS2021 [16] | ICISPC2019 [17] | ICET2020 [18] | Ours |
|---|---|---|---|---|
| FPGA Platform | ZC709 | XC7K325T | ZCU102 | VC709 |
| Target Network | YOLOv2-tiny | YOLOv2 | YOLOv2 | YOLOX-s |
| Precision/bit | Fix-8 | Fix-32 | Fix-16 | Fix-16 |
| Clock Frequency/MHz | 200 | 100 | 300 | 200 |
| DSP | 610/900 | — | 609 | 1024/1200 |
| BRAM | 256/545 | — | 491 | 1094/1470 |
| LUT | 84 K/219 K | — | 95 K | 165 K/433 K |
| FF | 65 K/437 K | — | 90 K | 228 K/866 K |
| Throughput/GOPS | 464.5 | 6.222 | 102.2 | 399.6 |
| Power/W | 10.25 | 2.555 | 11.8 | 14.76 |
| DSP efficiency | 95.2% | — | — | 97.56% |

## 4. Conclusions

Aiming to solve the problem of traditional remote sensing image processing methods not being able to efficiently and robustly handle satellite-to-ground target detection on satellite platforms with limited resources and space, this paper proposes and implements an efficient YOLOX-s inference acceleration architecture on an on-orbit FPGA. Combined with our proposed multi-level cache and three-path cache queue strategy, the efficient data reusing mode, small-scale cascade pooling, etc., the proposed architecture achieves an average throughput of 399.62 GOPS, and the computing efficiency of the entire DSP array is as high as 97.56%. Compared with previous research, our work further improves the

throughput and efficiency while ensuring a reliable design. For satellite-based platforms with limited resources and power consumption, our architecture design has great advantages and advancements. Additionally, our design provides a new direction for future research work on high-performance, high-efficiency, and compatible forward inference on satellite platforms.

**Author Contributions:** Conceptualization, methodology, software, formal analysis, investigation, data curation, and writing—review and editing, L.W.; validation, and writing—original draft preparation, K.J.; supervision, C.B.; project administration, and funding acquisition, H.Z.; resources, X.C. All authors have read and agreed to the published version of the manuscript.

**Funding:** This research was funded by the Youth Innovation Promotion Association of the Chinese Academy of Sciences Funding Project (Grant No. E0293401).

**Data Availability Statement:** Not applicable.

**Acknowledgments:** The authors would like to thank all the reviewers who participated in the review.

**Conflicts of Interest:** The authors declare no conflict of interest.

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
