# Peer review of "Hardware Acceleration and Implementation of YOLOX-s for On-Orbit FPGA"

_electronics, doi:10.3390/electronics11213473_

Round 1
Reviewer 1 Report
I really appreciated the content and quality of the presentation of the paper. The design flow and the hardware architecture are well explained. Also, the experimental results and comparison with other state of the art works are adequate. I have only some minor amendments that you can find in the attached annotated pdf.

Author Response
I would like to express my heartfelt thanks to the reviewers for your many modification suggestions, which are of great help to my article. I will try my best to modify the article based on the following suggestions.
Point 1: Higher than what?
Response 1: The higher speed and accuracy requirements are mainly oriented to more and more high-resolution remote sensing image data. We updated this description in the latest revision.
Point 2: Which FPGA device/board are you using? Data without the target device incomplete.
Response 2: We made it clear that the device is VC709 in the latest revision.
Point 3: Faster and higher accuracy if compared to what? On which datasets?
Response 3: The comparison here is a comparison with other YOLO series networks and single-stage networks such as EfficientDet on the public dataset COCO (We refer to the expression of the article cited here), We updated this description in the latest revision.
Point 4: BN, SiLU, CBS, CSP, and SPP are acronyms. It would be better to explicit them to help unexperienced readers.
Response 4: We have fixed the acronyms here in the latest revision.
Point 5: MIG stands for?
Response 5: Memory Interface Generator, We fixed the abbreviation in the latest revision
Point 6: Why DDR3? A lot of boards use DDR4? Are you already targeting a certain device at this point of the manuscript? If yes, why was not it mentioned?
Response 6: The FPGA chips we are currently using are not very compatible with DDR4. In addition, the main memory used on the current satellite is DDR2 and DDR3. As far as we know, DDR4 is very rarely used on satellites. We used a chip from the Micron MT8KTF51264HZ family, we updated this description in the latest revision (in section 3.1)
Point 7: power dissipation should be also given
Response 7: We have given the power consumption comparison in Table 3. At the same time, we have newly added a simple analysis of power consumption.
Point 8: Figures 16, 17 and 18 don't give any important information, please consider to remove them
Response 8: We present Figures 16-18 to illustrate table2 in more detail.

Reviewer 2 Report
This paper proposed an extremely efficient and reliable acceleration architecture for forward inference of the YOLOX-s detection network on-orbit FPGA to reduce the inference delay. And the experimental results demonstrated the feasibility of the proposed architecture. The paper is interesting and organized well. Some of the concerns are shown as follows to further refine the paper.
(1) In line 108, what is the Focus component?
(2) In lines 482-484, why do you claim the actual available DSP resources are 1200? Do you implement the TMR processing system together with the proposed accelerator in one FPGA chip?
(3) How much is the maximum clock frequency after implementation? Do you put constraints during synthesis and implementation?
(4) In Figures 16, 17, 18, are waveforms the function simulation or timing simulation? How do you emulate the DDR3? For example, how do you estimate the reading and writing from the DDR3 memory? How to calculate the DSP computing efficiency?
(5) In Figures 16, 17, 18, please explain why the actual computing time is larger than the ideal results? In Figures 16 and 18, the input size of the feature map is same, why the computing time is increased as the convolution kernel size is decreased?
(6) In general, the DSP computing efficiency denotes how many DSPs are involved in computing. In Equation (7), how to compute the throughput in your experiment? When data are 8-bit fixed-points, β is 4. I think this is not correct. In Virtex 7 FPGA, each DSP contains a 25x18 multiplier and an adder. When input data are less than 18-bit, they will utilize one DSP and operations are 2 (multiplication and addition) in maximum. Therefore, β is 2 when data are 8-bit and 16-bit.
(7) What is the real performance? You can input some real datasets to test the system. It is attractive and valuable to prove your proposal by showing such results.
(8) Minor revision of English is required to refine the paper. I am sorry I am not a native English speaker, please look for a native English speaker to refine it.
(a) In lines 9 and 10, “due to the limited hardware resources and space and power consumption constraints on the satellite” -> due to the limited hardware resources, space, and power consumption on the satellite.
(b) In lines 10 and 11, “ it is difficult for traditional remote 10 sensing image processing methods to process massive remote sensing images efficiently and 11 robustly.” à it is difficult to process massive remote sensing images efficiently and robustly using traditional remote sensing image processing methods
(c) In line 65, “None of the above-mentioned designs consider the …”, -> None of the above-mentioned designs considers….
(d) In line 481, “The clock 480 frequency of the FPGA is set to 200 MHz After synthesis…”
(e) In line 509, “the convolution kernel of size”
Author Response
I would like to express my heartfelt thanks to the reviewers for your many modification suggestions, which are of great help to my article. I will try my best to modify the article based on the following suggestions.
Our instructions may not be displayed completely on the webpage, please download the corresponding attachments.
Point 1: In line 108, what is the Focus component?
Response 1: The Focus component is a novel structure mainly composed of Slice operations (Figure 1), which can reduce the size of the feature map without losing the feature map information. We updated this description in the latest revision of Section 2.1.
Point 2: In lines 482-484, why do you claim the actual available DSP resources are 1200? Do you implement the TMR processing system together with the proposed accelerator in one FPGA chip?
Response 2: Yes, Our Design is Based on Single Board TMR Processing System. In the latest revision of Section 3.2, we have clearly revised the expression.
Point 3: How much is the maximum clock frequency after implementation? Do you put constraints during synthesis and implementation?
Response 3: We have added constraints. After final synthesis and implementation, the frequency can be up to 202.7Mhz. We have updated the description in the latest revision of Section 3.2.
Point 4: In Figures 16, 17, and 18, are waveforms the function simulation or timing simulation? How do you emulate the DDR3? For example, how do you estimate the reading and writing from the DDR3 memory? How to calculate the DSP computing efficiency?
Response 4:
- Function simulation and our implementation has met the timing constraints; We updated this description in the latest revision.
- In Vivado, with the help of the Memory Interface Generator(MIG) core, after we complete the parameter configuration of DDR3, we can get the MIG 7series instantiated entity. Based on the MIG 7series instance, the DDR3 parameter files wiredly.v, ddr3_model_parameters and the top-level testbench file sim_tb_top.v are generated under the IP Example Design. Based on the sim_tb_top.v, we instantiate the ddr3_model multiple times, so that we can perform DDR timing simulation.
- To realize the reading and writing of DDR3, we use MIG core to manage the read and write control timing of DDR3. On this basis, we also develop a further interface management module for the MIG core, which is the Memory Access Interface Control Module we mentioned in Figure 2.
- DSP computing efficiency is defined as the ratio of actual computing time to ideal computing time in convolution computing. The ideal computing time is only the time when the DSP array is performing computations to complete the current convolution computation at every CLK.
The actual time is the interval between loading the feature map from the DDR to calculating the entire feature map. The actual computing time includes the time that data is loaded from the memory to the DSP array and written back to the memory from the DSP array, and the time it takes for the DSP to perform the computation. In the latest revision of Section 3.2. we have refined the description of DSP computational efficiency.
Point 5: In Figures 16, 17, and 18, please explain why the actual computing time is larger than the ideal results. In Figures 16 and 18, the input size of the feature map is the same, why the computing time is increased as the convolution kernel size decreased?
Response 5: 1. The actual computing time includes DSP calculation, data loading time, and data output time. The ideal computing time is only the DSP computing time under the ideal bandwidth and does not consider the time of data loading and output, so it is less than the actual computing time anyway. At the same time, for a few special convolutional layers, such as Figures 17 and 18, due to the bottleneck of the bandwidth of the external DDR, the DSP will wait during the calculation process.
- For Figures 16 to 18, when the input size of the feature map is the same, as the size of the convolution kernel decreases, the computation time decreases, and the computing efficiency also decreases. The calculation time is reduced, mainly because the convolution kernel with a convolution of 1*1 has much less computation than a 3*3 convolution kernel. The computing efficiency is also reduced, because from Figure 16 to 18, the output channels of the convolution kernel are less and the kernel size is also reduced, so the reuse degree γ is reduced, so the bandwidth bottleneck of DDR appears, and the calculation efficiency of DSP is also reduced accordingly.
In the latest revision of Section 3.2, we have added a result analysis of Figures 16-18.
Point 6: In general, the DSP computing efficiency denotes how many DSPs are involved in computing. In Equation (7), how to compute the throughput in your experiment? When data are 8-bit fixed-points, β is 4. I think this is not correct. In Virtex 7 FPGA, each DSP contains a 25x18 multiplier and an adder. When input data are less than 18-bit, they will utilize one DSP, and operations are 2 (multiplication and addition) in maximum. Therefore, β is 2 when data are 8-bit and 16-bit.
Response 6: 1. , After simulation, we divided all convolutions into three categories based on computational efficiency, and got the forward inference time of all layers. In the latest revision of Section 3.2, We have reorganized the language expression here.
- Regarding the β question, I am very sorry to bring you the problem, in our article, Equation 7 is what we quoted from other articles, but I forgot to add the reference mark. In our work, we use FIX-16, so β=2. But for fix-8, some articles mentioned that β is 4, and some scholars proposed to use DSP to complete two 16-bit multiplications, of which 16 bits are two 8-bit precision numbers. Therefore, it is equivalent to a clock that can multiply two 8bit numbers in parallel. In this case, it is necessary to carry out a special carry design for the 16bit data, and also need to carry out other special designs for the input and output of the DSP. I've put an introduction to the citation here and the corresponding link, if there are any questions, we can have further discussions, thank you very much for such a wonderful question.
In the latest revision of Section 3.2, We have added a reference to Equation 7.
Citation link: A Low-Latency FPGA Implementation for Real-Time Object Detection | IEEE Conference Publication | IEEE Xplore
Point 7: What is the real performance? You can input some real datasets to test the system. It is attractive and valuable to prove your proposal by showing such results.
Response 7: Our current work is tested based on the generated image data. Our work mainly focuses on the design of the hardware architecture and the optimal design of the main computing modules in YOLOX-s, and finally completes the overall realization of the YOLOX-s network. If tested on real datasets, our system requires a complete image preprocessing system (sending real data), and the test efficiency is very low. At present, our project partners are mainly completing the development of other sub-systems.
Although the real data set is not used in the current test, the image data we generate is based on standard images for testing. This has no impact on assessing the functionality and performance of our architecture. Thank you from the bottom of my heart for your suggestion.
Point 8: Minor revision of English is required to refine the paper. I am sorry I am not a native English speaker, please look for a native English speaker to refine it.
Response 8: I am very grateful for the problems you found in my language expression. We have modified it based on your suggestion. In the latest revision, we have rechecked and corrected some of our previous language errors.
